# Crack Detection of FRP-Reinforced Concrete Beam Using Embedded Piezoceramic Smart Aggregates

**DOI:** 10.3390/s19091979

**Published:** 2019-04-27

**Authors:** Tianyong Jiang, Yue Hong, Junbo Zheng, Lei Wang, Haichang Gu

**Affiliations:** 1School of Civil Engineering, Changsha University of Science and Technology, Changsha 410114, China; hycsust@yeah.net (Y.H.); leiwang@csust.edu.cn (L.W.); 2Central-Southern Safety & Environment Technology Institute Co., Ltd., Wuhan 430051, China; junbozheng@yeah.net; 3Department of Mechanical Engineering, University of Houston, Houston, TX 77204, USA

**Keywords:** fiber-reinforced polymer (FRP) reinforced concrete beam, piezoceramic transducer, smart aggregates (SAs), concrete crack damage, wavelet packet energy

## Abstract

In this paper, the authors present a stress wave-based active sensing method to detect the crack in FRP-reinforced concrete beams. The embedded smart aggregates (SAs), which utilize Lead Zirconate Titanate (PZT) as transducers, are employed in this research to generate and sense the stress wave. Three specimens are involved in the experimental program and each is made of concrete, longitudinal distributed reinforcement, steel stirrups, main bar (FRP bar or steel bar), and four SAs. A pair of SAs installed on the lower part of the main bar and the other pair of SAs mounted on the upper part of main bar are utilized to monitor the crack occurrence and development in the three test specimens. The signals received by the SA sensors are analyzed in both time domain and frequency domain. The wavelet packet energy is used to extract damage features. The applied load–vertical displacement curves of mid-span in the specimen are obtained. Experimental results show the test specimens experience crushing failure when the concrete compression exceeds its compressive strength. Increasing the contact area between FRP bar and concrete can effectively improve the cracking load of the FRP-reinforced concrete beam and reduce the cracking speed and depth of FRP-reinforced concrete beam; on the other hand, increasing the elastic modulus of the main bar can slow down the crack development of concrete on the upper side of the main bar and decrease the displacement of reinforced concrete beam during the loading test process. The research results show that the developed piezoceramic-based active sensing method, though low-cost, can monitor the crack-induced damage and estimate the process of damage degree in real-time, and has potentials to provide an early warning of crack occurrence and development for FRP-reinforced concrete beams.

## 1. Introduction

Traditional steel reinforced concrete structures often suffer from corrosion [1,2,3]. On the other hand, fiber-reinforced polymer (FRP) bars have emerged as a novel alternative to conventional steel bars in concrete structures [4,5] because of their advantages of high corrosion resistance, high tensile strength, and magnetic neutrality [6,7]. However, owing to its low elastic modulus, more and wider cracks appear in the FRP-reinforced concrete beams, as compared with the steel reinforced concrete beams [8,9]. The durability and reliability of FRP-reinforced concrete structures will be negatively impacted with the occurrence and development of the cracks [6,10]. The structural members will experience severe damages when the crack width exceeds a certain value, which may lead to the concrete structures collapse. For the sake of reducing and avoiding economic losses caused by the crack damages, it is essential to develop efficient techniques to monitor and assess the crack damage of FRP-reinforced concrete beams [11,12].

Crack detection and monitoring in reinforced concrete structures has received much attention in the literature [13,14]. The current structural health monitoring (SHM) approach for crack monitoring mainly includes ultrasonic method, acoustic emission (AE), image processing, optical fiber sensor, coaxial cable sensor, and among others. The ultrasonic method is a conventional tool to detect cracks of civil engineering structures. Watanabe et al. employed the ultrasonic measurement to estimate the corrosion-induced cracks of concrete beam and the corrosion of rebar in two stages: dormant and acceleration [15]. Rucka and Wilde presented an ultrasonic wave to evaluate and monitor the distributed cracks in concrete by the PSD amplitudes as well as the frequency shifts and additional frequency component [16]. Seher et al. presented the theoretical background and numerical simulations of the diffuse ultrasound [17], and In et al. validated the diffuse ultrasonic method to measure the depth of a vertical crack in concrete beams [18]. Chen et al. examined that the fully noncontact second harmonic generation (SHG) technique using ultrasonic transducers was promising for the feasibility of detecting the crack characterization of concrete [19]. AE is one of the nondestructive evaluation (NDE) techniques that can be applied to monitor crack in concrete structures. Ohno et al. estimated the crack and degradation in concrete structures by AE techniques, and discussed two crack classification methods of the parameter analysis and the simplified Green’s functions for moment tensor analysis [20]. Luo et al. demonstrated the applicability of the AE technique to assess the damage level of railway structures via the AE indices [21]. Shiotani et al. investigated the crack or damage of concrete by AE detection and verified the ability of AE monitoring on the pile integrity [22]. Shahidan et al. used the AE analysis to determine crack movement and classify damage levels in the RC beam during an increasing cycle [23]. Nor et al. reported the fatigue crack damage monitoring to observe the crack patterns and classify the appearance of cracks on the reinforced concrete beam based on AE sensors [24]. Image processing can visualize the image of crack patterns [25]. Rouchier et al. developed a detection method of the simultaneous use of a digital image correlation and AE for monitoring the damage of concrete samples from microcracks to macroscopic fractures [26]. Nazmul et al. explained a method of image collection and analysis to measure crack opening displacements (CODs) on concrete surface by exploiting the Tikhonov regularization method [27]. Jahanshahi et al. presented a technique of multi-image stitching and scene reconstruction to evaluate change evolution in structures [28]. The sensing technique of optical fiber sensor, especially the distributed optical fiber (DOF) sensor, has been popularly used to detect the formation of structural cracks. Leung et al. developed a novel optical fiber sensor for crack detecting and monitoring [29]. Leung et al. also carried out the monitoring experiments of concrete beams with embedded DOF sensors [30]. Villalba et al. developed a promising measurement of optical backscatter reflectometer (OBR) sensors, which bonded to the concrete surface, to monitor and locate cracks [31]. In recent years, an innovative technique of coaxial cable sensor has been widely adopted for detecting cracks in concrete structures. Sun et al. utilized a novel coaxial cable sensor featuring time-domain reflectometry (TDR) sensing technology to capture the crack patterns in concrete structures [32]. Chen et al. implemented a distributed coaxial cable sensor of electrical time-domain reflectometry (ETDR) sensing to identify the crack appearance in a full-scale reinforced concrete girder [33]. Also, Chen et al. illustrated the utility of coaxial cable sensors with ETDR, which was employed to detect cracks or tensile strain in RC beams [34]. However, most of the above methods are not suitable for real-time application, often due to the high cost of the system and or the sensor probes. Therefore, it is necessary to develop a low-cost real-time method to monitor the crack occurrence and development of the reinforced concrete beam. To enable the real-time monitoring, the sensors have to be low-cost and can be integrated with a concrete structure and the data acquisition system shall have low-cost.

In recent years, due to their advantages, such as low-cost [35], wide bandwidth [36,37], dual sensing and actuating functions [38,39], energy harvesting [38,40], and strong piezoelectric effect, lead zirconate titanate (PZT) transducers have been increasingly researched structural health monitoring (SHM) [41,42,43]. The development of the smart aggregate (SA), whose key component is a protected PZT patch, enables embedment of PZT transducers in concrete structures for structural health monitoring [44,45,46]. Since then, SAs have been actively used for damage detection, including crack detection in concrete structures [47,48,49,50,51]. Feng et al. used an active sensing technique of SAs with wavelet packet energy analysis to evaluate the curing of concrete surface crack [52]. Song et al. identified crack existence inside the concrete structure employing the embedded PZT patch with wavelet packet analysis and a damage indicator [45]. Hughi et al. found that embedded piezoceramic sensors of harmonic wave propagation work well for measuring crack width and locating crack in reinforced concrete beams [53]. Jiang et al. presented the application of SAs with stress wave propagation monitoring corrosion-induced crack in prestressed concrete structure [54]. Dumoulin et al. developed an embedded PZT transducer for tracking crack growth in reinforced concrete beam based on time domain analysis and frequency domain analysis [55]. Zhu et al. verified the reliability and feasibility of PZT patches and SAs for estimating crack damage in concrete structures by carrying out the experiments [56]. Li et al. illustrated the ability of embedded SAs to evaluate the damage of the concrete structure, as compared with the surface mounted AE sensors [57]. As an alternative, the electromechanical impedance (EMI) has successfully received considerable attention in structural health monitoring [58,59,60,61], including detecting crack-induced damage in reinforced concrete structure [62,63,64]. Fan et al. proposed an EMI-based damage monitoring technique for evaluating impact damage of concrete column using root mean square deviation (RMSD) damage indicator [65]. Park et al. presented the feasibility and applicability of an impedance-based real-time health monitoring technique utilizing PZT patches for detecting crack in concrete structures [66]. Wang et al. proposed a new damage detection method based on electromechanical admittance (inverse of impedance) of multiple PZT patches and a damage index, namely, cross-correlation coefficient (CC). The numerical and experimental study results showed the detection technique can determine the damage locations of the plain concrete beam [67]. Li et al. developed an electromechanical impedance-based SHM technique of PZT patches for detecting the debonding damage of a CFRP rebar reinforced concrete [68].

In this paper, an active sensing technique based on wave propagation using SAs is developed to detect the occurrence and development of cracks in FRP-reinforced concrete beam. The test involves the three specimens with two pairs of SAs in each specimen. A pair of SAs are placed on the lower part of the main bar to monitor the cracking of the bottom of reinforced concrete beam. The other pair of SAs are installed on the upper part of the main bar to monitor the development of cracks after cracks pass through the main bar. The dial indicators are installed at the mid-span and both ends of the test specimens. A screw jack is used to apply the load to the test specimens through a distribution beam, and a load cell is used to measure the force. In the three test specimens, the experimental program consists of two different experiments parameters, such as the contact area between main bar and concrete, and the elastic modulus of main bar. The cracks attenuated the propagating stress wave energy which can be reflected by the received signal in time domain, frequency domain, and the wavelet packet-based energy index. The experimental research validates the feasibility of monitoring the cracking in FRP-reinforced concrete beam using SA sensors based on active sensing technique.

## 2. Crack Detection Principle

### 2.1. An Active Sensing Method

A three-dimensional diagram of cracking detection in reinforced concrete beams is shown in Figure 1. In this research, the concrete cracks often begin to appear at the lower part of the reinforced concrete beam, and then gradually develop upwards as the load increases, through the main bar, and until the test specimen fails. In order to monitor the cracking occurrence and development, two pairs of SAs are installed on the lower and upper part of the main bar in reinforced concrete beam, respectively. Each pair of SAs form an actuator and a sensor pair, wherein the actuator is used for generating a stress wave and the sensor is utilized to detect the response, as shown in Figure 1. The excitation frequency for the active sensing methods is normally less than 200 kHz, which is much less than that of an impedance method or AE method [12,58,59], lowering the cost of the entire system.

The principle of the active sensing approach is illustrated in Figure 2. When the reinforced concrete beam is in a healthy state without cracks before loading, the sensors can receive the relatively strong stress signals, as shown in Figure 2a. With the load increasing, tensile stress appears first at the lower part of the reinforced concrete beam. When the tensile stress exceeds its tensile strength, the reinforced concrete beam begins to crack. At this time, as the incident wave may be reflected in the concrete propagation process, the smart aggregate SA3 receives fewer stress wave signals, as shown in Figure 2b. As the load increases further, cracks develop upward in the reinforced concrete beam, even passing through the main bar. This will cause that the signals received by the sensor SA3 installed on the lower part of the main bar are very small, and the signals received by the sensor SA4 installed on the upper part of the main bar are also reduced, as shown in Figure 2c. When the reinforced concrete beam is destroyed, the width of the cracks near the main bar is very large, resulting in a very high reflection wave energy, which makes the number of signals available to be received very small, so much so that it is almost negligible, as shown in Figure 2d.

### 2.2. Wavelet Packet-Based Energy Analysis

Obviously, the development of cracks can be investigated through the variation of stress wave received by the smart aggregate sensors in the reinforced concrete beam. Cracking should be monitored and quantitatively evaluated throughout the test loading process. The wavelet packet-based energy analysis has been widely applied in structural health monitoring in recent years [45,46,47], and can quantitatively illustrate structural damage in the reinforced concrete beams. In the wavelet packet-based energy analysis, the sensor signal can be decomposed by *n*-level wavelet packet decomposition into 2*^n^* frequency bands. The signal energy of each frequency band can be computed by the summation of the square of each sampling data in the frequency band. Furthermore, the total of the signal energy is obtained by the summation of the signal energies from all frequency bands. In this research, the wavelet packet-based energy analysis is used to quantify the energy of the signals received by the SA sensors. The energy index of the received signals can be established for the purpose of directly comparing the change of the received signal energy during the test loading process. In the wavelet packet-based analysis, the sensor signal *X* can be decomposed by a *n*-level wavelet packet decomposition into 2*^n^* frequency bands. *X_j_* can be expressed as
*X_j_* = [*X*_*j*,1_, *X*_*j*,2_, …, *X*_*j*,*m*_](1)
where *m* is the number of sampling data and *j* is the frequency band (*j* = 1, 2, 3, …, 2*^n^*). Additionally, the total energy of the decomposed signal *E*_*i*,*j*_ can be defined as
*E*_*i*,*j*_ = ||*X_j_*||^2^ = *X*_*j*,1_^2^ + *X*_*j*,2_^2^ + … + *X*_*j*,*m*_^2^(2)
where *i* represents the data measured in different time. The total signal energy at time *i* can be computed by the summation of the decomposed signal energy *E_i_*
*E_i_* = [*E*_*i*,1_, *E*_*i*,2_, …, *E*_*i*,2_^*n*^](3)

In the initial state, the healthy signal energy (*E_h_*) is measured, and the signal energy collected is recorded as *E_i_* at different times. During the loading test, cracks begin to appear at the bottom of the reinforced concrete beams. When the stress wave encounters a crack, there will be an obvious reflection wave and a weakened received wave. The cracks will continue to develop with the load increasing, the signal energy *E_i_* received by the SA sensor will continuously decrease. When the reinforced concrete beam is completely destroyed, it will make the crack width so large that the SA sensor can hardly receive any signal and the wavelet packet energy will be almost zero.

## 3. Experimental Investigation

### 3.1. Specimen Details

To monitor the cracking characteristics of reinforced concrete beams, especially FRP-reinforced concrete beams, three test specimens are designed with a calculated span of 1800 mm and a total length of 2000 mm, as shown in Figure 3. The test specimens involve mainly C40 concrete, main bar (FRP bar or steel bar), distributed reinforcement, steel stirrups, and SAs. In this experimental process, there are three test specimens—Beam A, Beam B, and Beam C—with a rectangular cross-section of 130 mm × 150 mm, as shown in Figure 4. Each specimen has different main bars: Beam A has one FRP bar with a diameter of 10 mm, as shown in Figure 4a; Beam B has four FRP bars with a diameter of 5 mm and the spacing of 16 mm, as shown in Figure 4b; and Beam C has one steel bar with a diameter of 10 mm, as shown in Figure 4c. The distance between the center of the main bar and the bottom of the beam is 55 mm.

In the research, each specimen has longitudinal distributed reinforcements with a diameter of 8 mm and steel stirrups with a diameter of 8 mm. The steel stirrups are arranged at different spacing along the length of the test specimen. The spacing of steel stirrups within the middle-span of ~600 mm is 100 mm, while the spacing of steel stirrups at other locations is 50 mm. In addition, the protective layer thickness of the steel stirrups is 25 mm, that is to say, the minimum distance between the outer edge of the steel stirrups and the test specimen is 25 mm, as shown in Figure 4. For each specimen, two pairs of SAs are installed on the lower or upper part of the main bar (FRP bar or steel bar) in the test specimen, respectively. The distance between the bottom of the test specimen and the edge of the sensors SA1 and SA3 installed on the lower part of the main bar is 15 mm, while the distance between the bottom of the test specimen and the edge of sensors SA2 and SA4 installed on the upper part of the main bar is 70 mm. The SAs are made by sandwiching the wire-leaded PZT patch between two marble blocks using epoxy resin. The type of the PZT patches used in the SAs is PZT-5H. The dimensions of the SAs are 25 mm × 25 mm × 25 mm. The mixture ratio of the concrete is shown in Table 1. The type 32.5 Portland cement is utilized. During the casting of concrete beams, three cubic test blocks are reserved. After standard curing for 28 days, the average compressive strength of the test blocks is about 40 MPa. The ultimate tensile strength of FRP bar is 1200 Mpa and the elastic modulus of FRP bar is 160 GPa. The ultimate tensile strength of the steel bar and the steel stirrup is 330 Mpa and the elastic modulus of the steel bar and the steel stirrup is 200 GPa [69,70].

Beam A and Beam B have the same elastic modulus of FRP main bar but different contact area between the main bar and the concrete, while Beam A and Beam C have the same contact area between the main bar and the concrete but different elastic modulus of main bar. Therefore, the experimental program involves two different comparative experiment stages: (1) monitoring the cracking characteristics of FRP-reinforced concrete beams with the same main bar but different contact areas between main bar and concrete, such as Beam A and Beam B, and (2) monitoring the cracking characteristics of FRP-reinforced concrete beam and steel reinforced concrete beam with same concrete contact area but different elastic modulus of main bar, such as Beam A and Beam C.

### 3.2. Experimental Setup and Procedures

The experimental setup includes the reinforced concrete test specimen with four SAs, one reaction frame, one load cell, one screw jack for loading the specimen, one concrete pad, one load distribution beam, two fixed supports and two sliding supports, three dial indicators for measuring the vertical displacement of the test specimen, two concrete support piers, one multifunctional strain measuring instrument for reading the value of the force sensor, one data acquisition board NI-USB 6363 with the function of generating and collecting the signals, and one supported laptop, as shown in Figure 5. The reinforced concrete test specimen with four SAs is mounted on the concrete support piers through the fixed support and the sliding support. The loads generated by the screw jack are transferred to the test specimen through the concrete pad and the load distribution beam with one fixed support and one sliding support. The reaction frame and the concrete support piers are installed on a strong floor. During the installation process of the test loading device, the center of the force sensor, the screw jack, the concrete pad, and the load distribution beam should be overlapped with the center of the reinforced concrete test specimen as far as possible, so as to ensure that the applied load will not have eccentric effect, and also to meet the effectiveness and safety of the test loading.

Detailed layout of the experimental loading is shown in Figure 6. The distance between the two loading points of the reinforced concrete test specimen is 600 mm. The distance between the two loading points and the support center of the reinforced concrete test specimen also is 600 mm. In order to monitor the vertical displacement of the reinforced concrete beam specimen, the dial indicators D1 and D2 are mounted on the top of the support of the test specimen, while the dial indicator D3 is installed at the mid-span bottom of the test specimen. The two pairs of SAs are mounted under the loading point of the reinforced concrete beam. Among them, SA1 and SA3 are installed on the lower part of the main bar with the monitoring distance of 600 mm; SA2 and SA4 are installed on the upper part of the main bar with the monitoring distance of 600 mm.

Before the formal loading test, to eliminate the initial defect of the test specimen and to ensure that the components of the specimen will be in good condition and the loading equipment and test instruments will work normally, the preloading test is carried out. During the loading process, the load applied by the screw jack is continuously applied to the test specimen through the load distribution beam. And the load is measured by the load cell and read by connecting the multifunctional strain measuring instrument. The static loading test is applied from zero until the failure occurs. The increment of each load class in the test is 2 kN. The vertical displacements of the test specimen are measured by the dial indicators under the different load levels in the static loading. At the same time, a steady swept sine wave signal is utilized as the excitation signal of embedded SA1 and SA2 actuators controlled by NI-USB 6363. The frequency range of the swept sine wave signal is from 100 Hz to 150 kHz. The amplitude and the period of the swept sine wave are 10 V and 1 s, respectively. The predetermined input signal (excitation signal) is directly programmed and sent to the SA1 and SA2 transducers, as actuators, from the NI-USB 6363. Since the data acquisition board NI-USB 6363 has the function of transmitting and receiving the signals simultaneously, the output signals from SA3 and SA4 transducers, as sensors, can be recorded utilizing the NI-USB 6363 in the loading test. After the data processing and analysis, the structure stiffness, time-domain signal, frequency domain and the wavelet packet-based energy index of the three test specimens are obtained.

## 4. Experimental Results and Discussions

### 4.1. Failure Photos of the Test Specimens

Failure photos of the test specimens are shown in Figure 7. From these failure photos of the test specimens, it can be seen that the three test specimens show that the concrete at the top of the test specimen appears crushing failure when the concrete compression exceeds its compressive strength. Among them, the degree of concrete crushing failure of Beam A is the largest, while the degree of crushing failure of Beams B and C is relatively small. Correspondingly, the results of the specimen cracking show that Beam A has relatively more cracks, while Beams B and Beam C have relatively fewer cracks. In particular, Beam C has the least cracks. Most of the cracks appear between loading points of ~600 mm in the test specimens.

By comparing the test failure photos of Beam A with one FRP bar with a diameter of 10 mm and Beam B with four FRP bars with a diameter of 5 mm, it can be seen that the degree of concrete crushing failure of Beam A is more severe than that of Beam B, and the crack length of Beam A is larger than that of Beam B, as shown in Figure 7a,b. This is mainly because the contact area between the main bar and the concrete of Beam B is twice that of Beam A. This indicates that the increase of the contact area between the main bar and the concrete improves the stress distribution of concrete, slows down the appearance and expansion of cracks, and improves the service performance of the reinforced concrete beams. For Beam A with one FRP bar with a diameter of 10 mm and Beam C with one steel bar with a diameter of 10 mm, the two test specimens have the same contact area between the main bar and the concrete, however the main bars have different elastic modulus of materials, that is to say, the elastic modulus of FRP bar is smaller, while the elastic modulus of steel bars is larger. When Beam A and Beam C are applied with the same load, the FRP-reinforced concrete beam experiences a greater displacement and a more severe crack than those of the steel reinforced concrete beam, as shown in the Figure 7a,c.

### 4.2. Stiffness Characteristics of the Specimens

Since the measuring points D1 and D2 are installed at the top of the supports, they are only used to measure the displacements of the supports under the applied load. And the measuring point D3 is mounted in the mid-span of the specimen to monitor the maximum displacement of the test specimen, thus it can be seen that the applied load–vertical displacement curves of the measuring point D3 shows the stiffness characteristics of the specimens. According to the relationship of the displacement of measuring points of the test specimen, the displacement of the measuring point D3 used in the following analysis should subtract the average displacement of the measuring point D1 and D2. The applied load–vertical displacement curves of the measuring point D3 of the test specimens are shown in Figure 8.

At the beginning of the applied load, the test specimens work in the elastic stage without cracking, and the vertical displacements of the measuring point D3 increase linearly with the applied load increasing. When the corresponding applied load arrives at the cracking load, the cracks in the test specimens begin to appear in the mid-span of the specimen. This will lead to the decrease of the stiffness of the reinforced concrete beam specimens, and the increase of the vertical displacement of the specimens under the same applied load, that is, the applied load–vertical displacement curves appear an obvious change at this cracking load. In the Figure 8, it can be seen that the applied load–vertical displacement curves of Beam A and Beam C have an obvious change in slope at the applied load of 2 kN; while the curve of Beam B has the obvious change in slope at the applied load of 4 kN. Therefore, the cracking load of Beam A and Beam C is 2 kN, while the cracking load of Beam B is 4 kN. This shows that the cracking load of Beam B designed four FRP bars with a diameter of 5 mm is larger than that of Beam A designed one FRP bar with a diameter of 10 mm. This indicates that increasing the contact area between FRP bar and concrete can effectively restrain the cracking of the concrete, which as a result increases the cracking load of the FRP-reinforced concrete beam and improves the mechanical performance of the test specimen.

For Beam A and Beam C, the curves of the mid-span applied load vs. the vertical displacement are similar. When the test specimens are applied with the same applied load, the vertical displacements of the Beam A (designed with one FRP bar with a diameter of 10 mm) are slightly larger than that of the Beam B (designed with one steel bar with a diameter of 10 mm). This is mainly because the FRP bar and steel bar have different elastic modulus, and the elastic modulus of FRP bar is lower than that of steel bar. Therefore, the stiffness of FRP-reinforced concrete beam is lower than that of reinforced concrete beams, which will lead to the greater displacement of FRP-reinforced concrete beam under the same applied load than that of steel reinforced concrete beam.

### 4.3. Time Domain Analysis

The time domain signals received by SA3 and SA4 sensors of Beam A are shown in Figure 9 and Figure 10, respectively. The time domain signals received by SA3 and SA4 sensors of Beam B are shown in Figure 11 and Figure 12, respectively. The time domain signals received by SA3 and SA4 sensors of Beam C are shown in Figure 13 and Figure 14, respectively. As the ultimate loads of Beam A and Beam C are 28 kN, their loading is divided into 14 working conditions. In order to reduce the space of this paper, the time-domain signal analysis only takes the signals of four typical working conditions: (a) 0 kN at the initial stage of loading, (b)10 kN at ~1/3 of the ultimate load, (c) 20 kN at ~2/3 of the ultimate load, and (d) 28 kN at the ultimate load. As the ultimate load of Beam B is 34 kN, the loading is divided into 17 working conditions. To reduce the length of this paper, the time-domain signal analysis also only takes the signals of four typical working conditions: (a) 0 kN at the initial stage of loading, (b)12 kN at ~1/3 of the ultimate load, (c) 22 kN at ~2/3 of the ultimate load, and (d) 34 kN at the ultimate load. The results in Figure 9, Figure 10, Figure 11, Figure 12, Figure 13 and Figure 14 show that the amplitudes of the voltage signal by the sensors decrease with the increase of the applied load in the test specimens. The reason is that the test specimens will crack when the applied load reaches the cracking load, and in the same time, the depth, width, and number of cracks will also increase with the increase of the applied load. As a result, more and more of the stress wave signals are reflected by the cracks, fewer and fewer stress wave signals can pass through the areas with cracks.

### 4.4. Frequency Domain Analysis

Compared to the time domain signals, the difference in the trend of the power spectra density (PSD) energy can be more easily observed from the frequency domain. The frequency domain signals received by SA3 and SA4 sensors of Beam A are shown in Figure 15 and Figure 16, respectively. The frequency domain signals received by SA3 and SA4 sensors of Beam B are shown in Figure 17 and Figure 18, respectively. The frequency domain signals received by SA3 and SA4 sensors of Beam C are shown in Figure 19 and Figure 20, respectively. To further reduce the length of this paper, based on the time domain analysis, the frequency domain analysis is only taken three working conditions: the initial loading, 1/3 of the ultimate load, and the ultimate load, as shown from Figure 15, Figure 16, Figure 17, Figure 18, Figure 19 and Figure 20. From these frequencies domain signals, the results show that the sensitive frequency range of the stress wave signal is from 35 to 50 kHz. In other words, the maximum frequency domain amplitude of the stress wave signal occurs in the sensitive frequency range. By comparing the amplitude of the signals in the frequency domain, it can be clearly seen that the amplitude of the signals decreases with the increase of the applied load in the test specimens, and the signal amplitude of the sensor SA3 installed on the lower side of the main bar in the test specimens decreases faster than that of the sensor SA4 installed on the upper side of the main bar in the test specimens. This is mainly because the test specimens cracking begins from the lower part of the reinforced concrete beam under the applied load, and gradually develop toward the upper part of the test specimens with the increase of applied load. Hence, under the same applied load, the damage degree of the concrete under the main bar in the test specimens is greater than that of the concrete on the main bar. Therefore, the sensors SA3 placed on the lower side of the main bar first can sense the signal degradation due to the concrete cracking, and the signal degradation degree of the sensors SA3 is also greater than that of the sensors SA4 installed on the upper side of the main bar in the test specimens.

### 4.5. Wavelet Packet Energy Analysis

To quantify the energy of the stress wave signals detected during the applied load process, the signal energy is analyzed by using the wavelet packet energy. The results are shown in Figure 21, Figure 22 and Figure 23. Figure 21 shows the energy levels of the sensors of Beam A during the applied load process. When the applied load arrives (2 kN), only in the first level load is the computed energy of the sensor SA3 installed on the lower side of the main bar reduced by ~72.7%, as shown in Figure 21a. While the computed energy of the sensor SA4 installed on the upper side of the main bar is also reduced by ~9.1% at the applied load of 2 kN, as shown in Figure 21b. This shows that when the applied load is only 2 kN, the cracking degree of the concrete at the lower part of Beam A is large, resulting in a great reduction in the stress wave signal energy received by sensor SA3. Moreover, the cracking has passed through the main bar and reached to the sensor SA4, resulting in a small decrease in the signal of sensor SA4. As the applied load continues to increase, the displacement of the test specimen increases and the crack of the reinforced concrete beam further develops, when the applied load is up to 10 kN, the signal energy received by sensor SA3 is very small, almost negligible. In addition, the stress wave signal energy received by the sensor SA4, when the applied load reaches at 4 kN, decreases the fastest, reaching 84.5%. Subsequently, the energy received by sensor SA4 continues to decrease until the end of the test experiment. However, it still receives a weak signal, and the energy is ~1.4% of the energy at the healthy state.

The wavelet packet energy analysis of the sensors of Beam B is shown in Figure 22. By comparing the wavelet packet energy of the sensors of Beam A and Beam B, it can be seen that the energy of Beam B with more contact area between the main bar and the concrete decreases more slowly with the increase of applied load. The stress wave signal energy received by the sensor SA3 of Beam B decreases the fastest at the applied load of 6 kN, while the energy received by the sensor SA3 of Beam A decreases the fastest at the applied load of 2 kN. This indicates that more contact area can delay the cracking time of concrete on the lower side of the main bar and improve the cracking load of concrete in the test specimen. For the sensor SA4 installed on the upper side of the main bar, the stress wave signal energy of Beam A decreases the fastest at the applied load of 4 kN, while the stress wave signal energy of Beam B reduces slowly, and the obvious energy can be received until the end of the test. In the loading process of Beam B, the slight cracks appear at the applied load of 4 kN, and the most significant crack is 0.02 mm in width and 4.5 cm in length. As the applied load continues to increase, the cracks also continue to extend with a width of 0.05 mm and a length of 7.5 cm at the applied load of 6 kN. Although the length of the crack has exceeded the height of the main bar, there is no sudden decrease in the wavelet packet energy, which can be illustrated that the concrete crack only extends on the surface of the concrete, but not to the interior. Therefore, increasing the contact area between the main bar and the concrete improves the cracking load and reduces the cracking speed and depth of FRP-reinforced concrete beam.

Figure 23 shows wavelet packet energy analysis of the sensors of Beam C. Taking Figure 8, Figure 21 and Figure 23 into consideration, Beam A and Beam C, with the same contact area between main bar and concrete, not only have similar mid-span applied load—vertical displacement curves, but also have similar analysis curves of wavelet packet energy of the sensors utilized for receiving the stress wave signals. The difference between Beam A and Beam C is that the main bar of Beam A is FRP bar with the elastic modulus of 160 GPa, while the main bar of Beam C is a steel bar with an elastic modulus of 200 GPa. By comparing Figure 21a and Figure 23a, it can be seen that the wavelet packet energy of the sensor SA3 installed on the lower side of the main bar of the two test specimens decreases the fastest at the applied load of 2 kN. The results show that the elastic modulus of main bar has little effect on the cracking load of the reinforced concrete beam. But by comparing Figure 21b and Figure 23b, we can see that the wavelet packet energy received the sensor SA4 installed on the upper side of the main bar of Beam A decreases by 84.5% at the applied load of 6 kN, while that of Beam C decreases by 66.2% at the applied load of 6 kN. This illustrates increasing the elastic modulus of the main bar can slow down the crack development of concrete on the upper side of the main bar in the test specimens.

## 5. Conclusions

This paper presents an experimental investigation of detecting crack-induced damage using a stress wave technique with embedded smart aggregates through three reinforced concrete beam specimens. Experimental results show that the crack-induced damages in FRP-reinforced concrete beam are successfully monitored by using the stress wave-based active sensing approach during the static load test in real-time. Experimental results show that the time domain amplitudes, frequency domain amplitudes and wavelet packet energies of the signal received by SA sensors decrease when the crack-induced damage occurs. With the increase of the applied load, the crack-induced damage increases gradually, resulting in that the transmitting stress wave signals are attenuated, and the signal amplitude of the sensors installed on the lower side of the main bar decreases faster than that of the sensors installed on the upper side of the main bar. The analysis results show that increasing the contact area between FRP bar and concrete can effectively increase the cracking load of the FRP-reinforced concrete beam and reduce the cracking speed and depth of FRP-reinforced concrete beam; increasing the elastic modulus of the main bar can slow down the crack development of concrete on the upper side of the main bar and decrease the displacement of reinforced concrete beam during the loading test process. Therefore, the developed piezoceramic-based active sensing method can monitor the crack occurrence and development for FRP-reinforced concrete beams in real-time. Since the appearance of crack in the FRP reinforcement concrete beams is random and uncertain, the author’s future work will involve the method to locate and quantify the spatial attitude of the cracking of FRP-reinforced concrete beams, which can accurately evaluate the damage degree of FRP reinforcement concrete beams caused by cracks. The piezoceramic smart aggregates are economical with low manufacturing cost and they can be easily integrated with concrete structures for real-time structural health monitoring. In addition, the low sampling rate requirement for the data analysis reduces the cost of the data acquisition system. These factors will pave the way for field implementation of the proposed method to monitor cracks in FRP concrete structures.

## Figures and Tables

**Figure 1 sensors-19-01979-f001:**
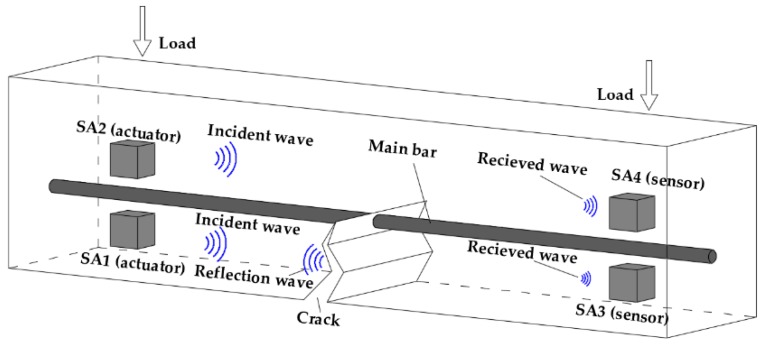
Three-dimensional diagram of cracking detection of reinforced concrete beams.

**Figure 2 sensors-19-01979-f002:**
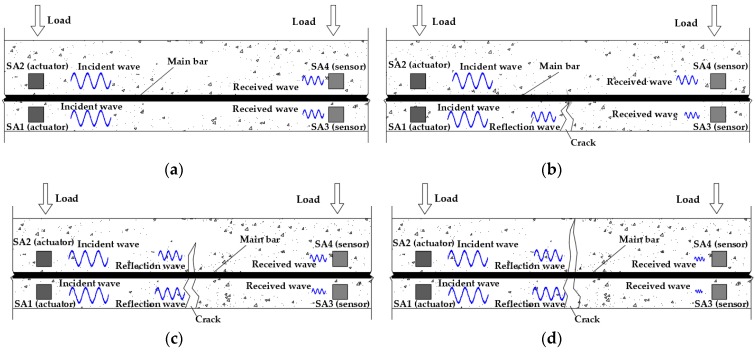
The diagrams to illustrate the active sensing approach. (**a**) The initial state before loading; (**b**) cracks occur under the main bar; (**c**) cracks grow and pass through the main bar; (**d**) and cracks develop until the reinforced concrete beam is destroyed.

**Figure 3 sensors-19-01979-f003:**
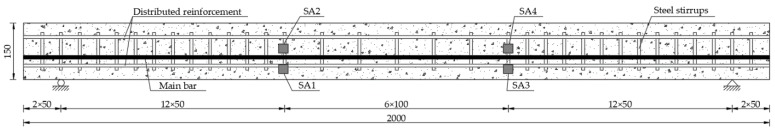
Test beam details (unit: mm).

**Figure 4 sensors-19-01979-f004:**
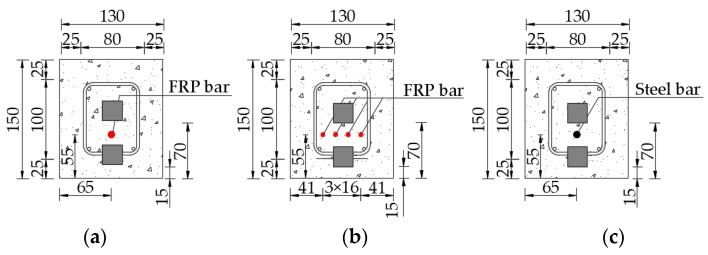
Cross-sections of three test specimens (unit: mm): (**a**) Beam A, (**b**) Beam B, (**c**) and Beam C.

**Figure 5 sensors-19-01979-f005:**
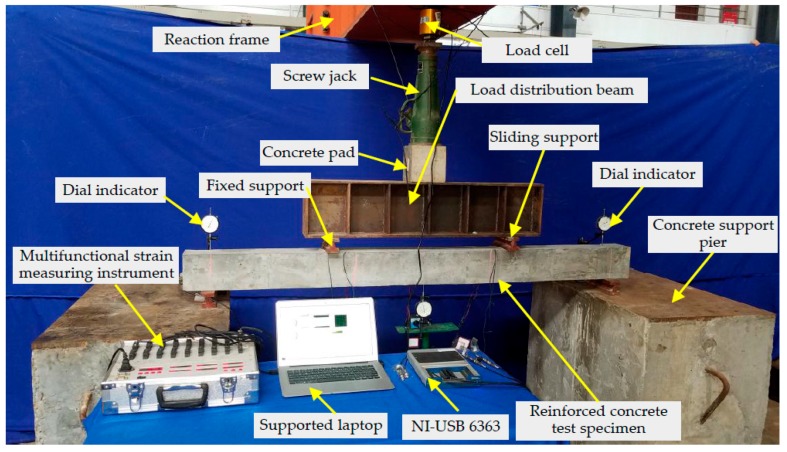
Photo of the experimental setup.

**Figure 6 sensors-19-01979-f006:**
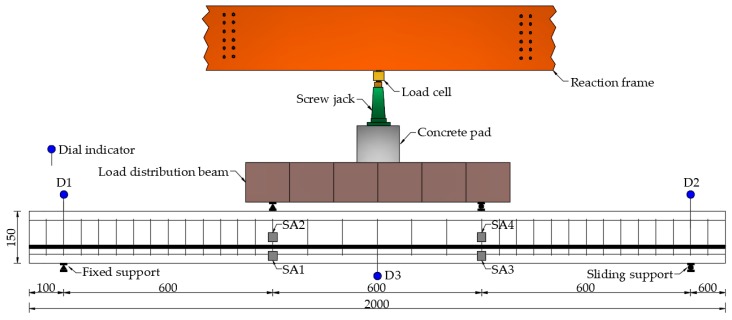
Detailed layout of the experimental loading system (unit: mm).

**Figure 7 sensors-19-01979-f007:**
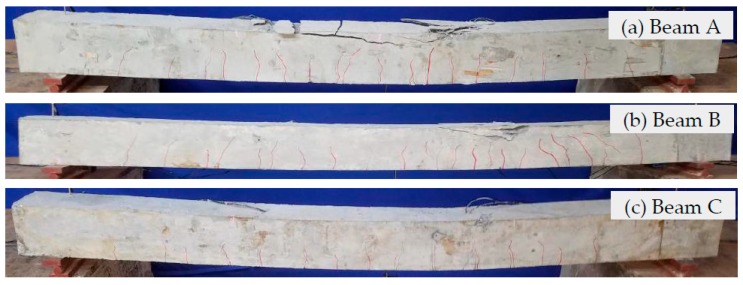
Failure photos of the test specimens.

**Figure 8 sensors-19-01979-f008:**
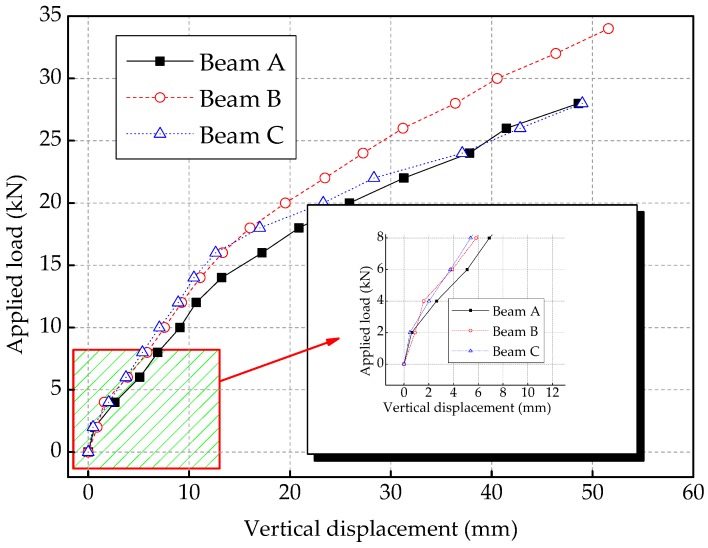
The applied load–vertical displacement curves of the measuring point D3 of the test specimens.

**Figure 9 sensors-19-01979-f009:**
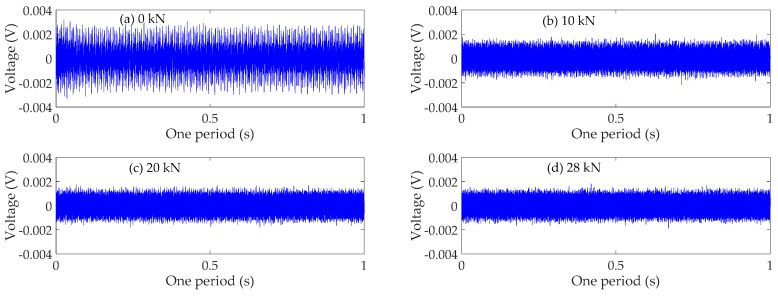
Time domain signals of the sensor SA3 of Beam A.

**Figure 10 sensors-19-01979-f010:**
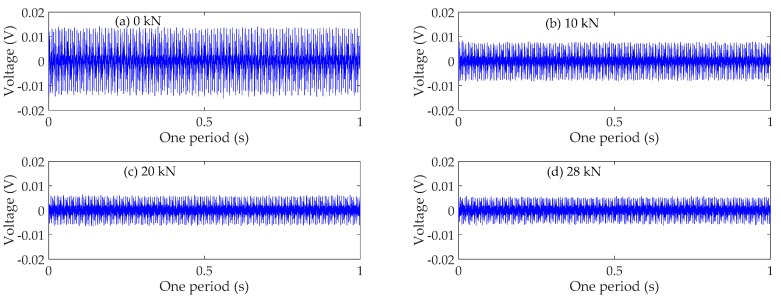
Time domain signals of the sensor SA4 of Beam A.

**Figure 11 sensors-19-01979-f011:**
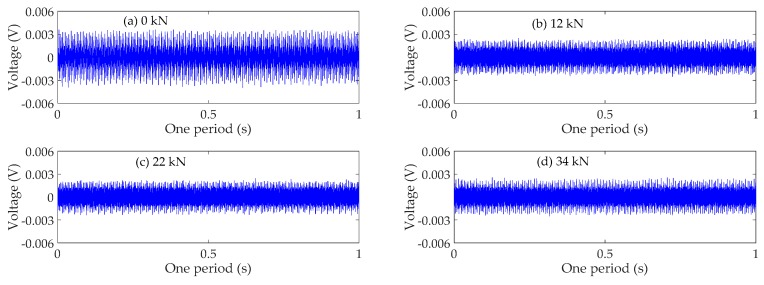
Time domain signals of the sensor SA3 of Beam B.

**Figure 12 sensors-19-01979-f012:**
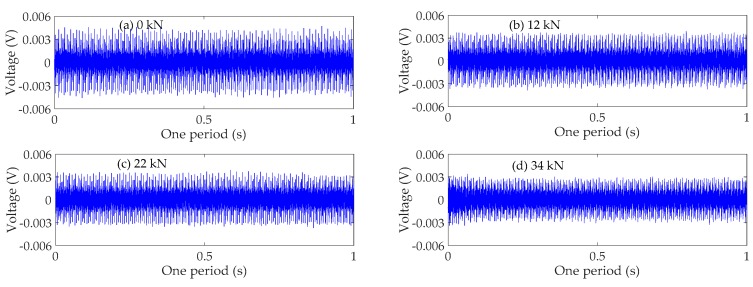
Time domain signals of the sensor SA4 of Beam B.

**Figure 13 sensors-19-01979-f013:**
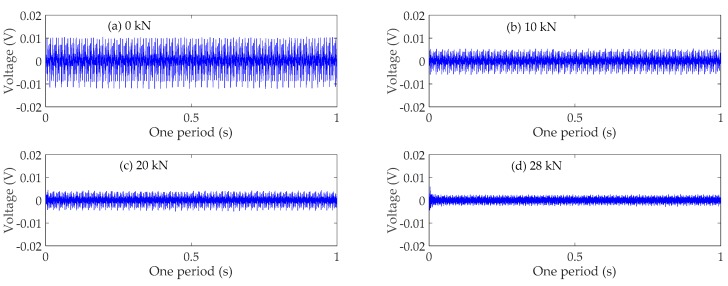
Time domain signals of the sensor SA3 of Beam C.

**Figure 14 sensors-19-01979-f014:**
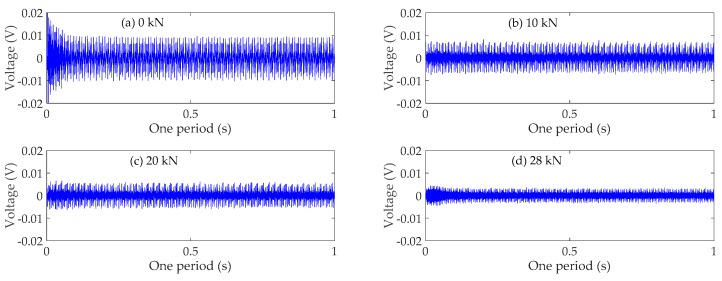
Time domain signals of the sensor SA4 of Beam C.

**Figure 15 sensors-19-01979-f015:**
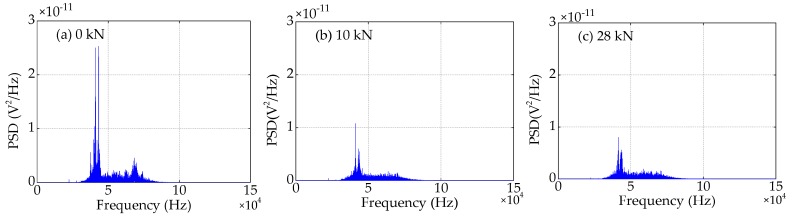
Frequency domain signals of the sensor SA3 of Beam A.

**Figure 16 sensors-19-01979-f016:**
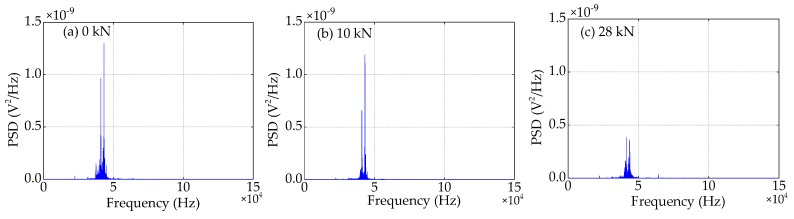
Frequency domain signals of the sensor SA4 of Beam A.

**Figure 17 sensors-19-01979-f017:**
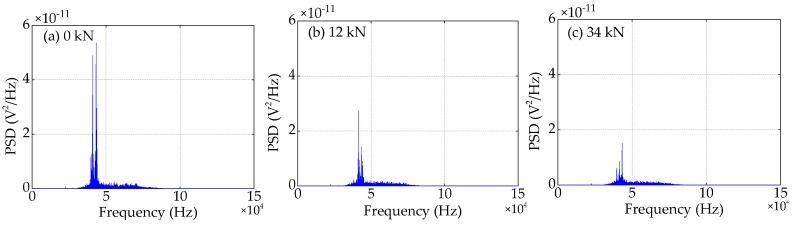
Frequency domain signals of the sensor SA3 of Beam B.

**Figure 18 sensors-19-01979-f018:**
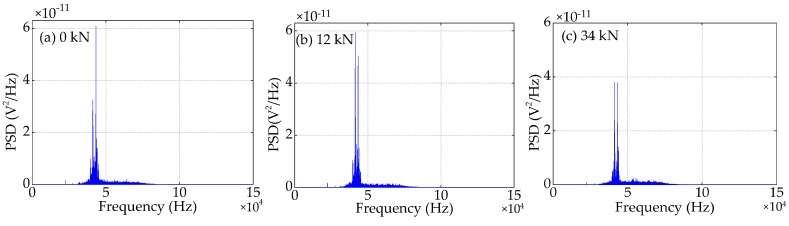
Frequency domain signals of the sensor SA4 of Beam B.

**Figure 19 sensors-19-01979-f019:**
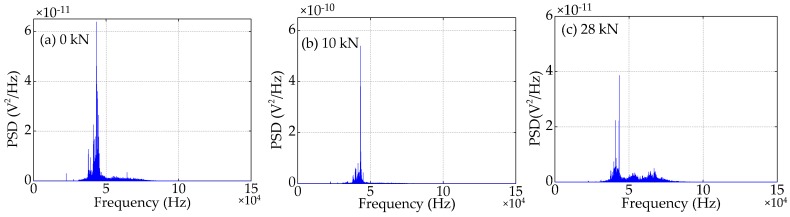
Frequency domain signals of the sensor SA3 of Beam C.

**Figure 20 sensors-19-01979-f020:**
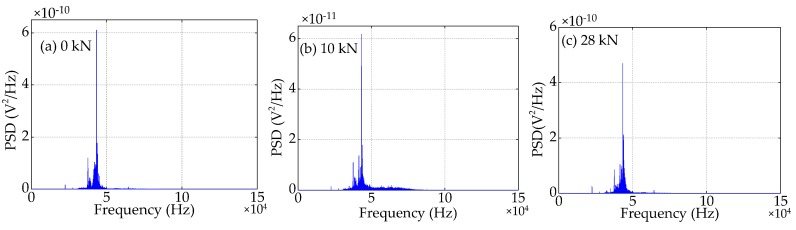
Frequency domain signals of the sensor SA4 of Beam C.

**Figure 21 sensors-19-01979-f021:**
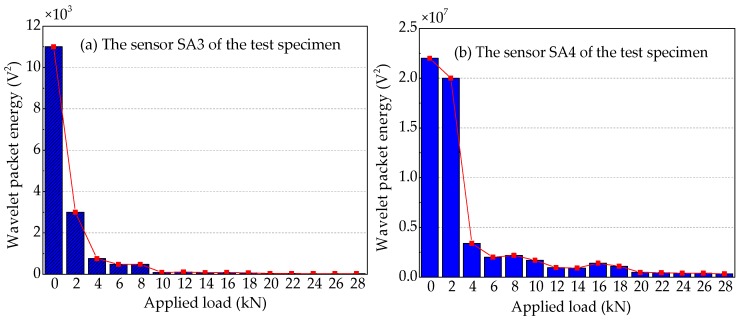
Wavelet packet energy analysis of the sensors of Beam A.

**Figure 22 sensors-19-01979-f022:**
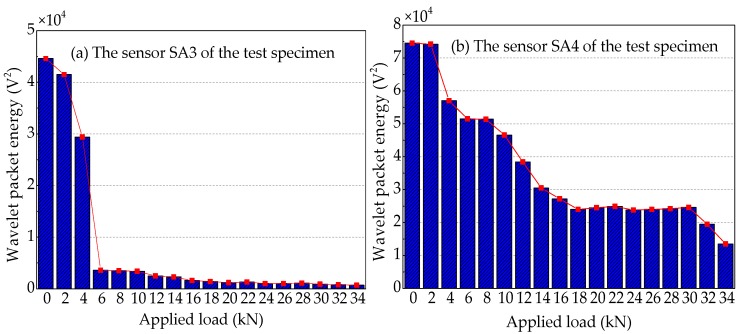
Wavelet packet energy analysis of the sensors of Beam B.

**Figure 23 sensors-19-01979-f023:**
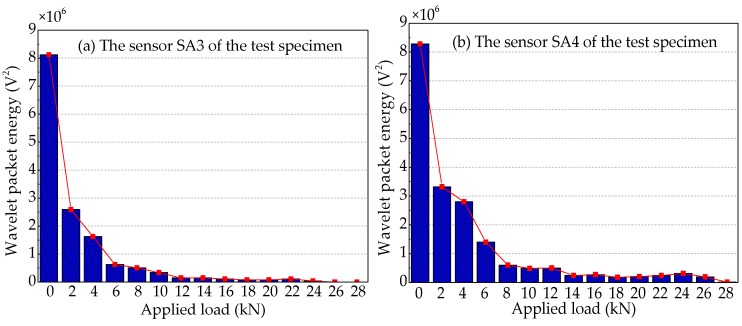
Wavelet packet energy analysis of the sensors of Beam C.

**Table 1 sensors-19-01979-t001:** Mixture ratio of the concrete.

Cement (kg/m^3^)	Water (kg/m^3^)	Sand (kg/m^3^)	Stone (kg/m^3^)	Water Reducing Agent (kg/m^3^)
444	165	757	1004	9.8

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
