# Peer review of "Crack Detection of FRP-Reinforced Concrete Beam Using Embedded Piezoceramic Smart Aggregates"

_sensors, 2019, doi:10.3390/s19091979_

Round 1

Reviewer 1 Report

This paper investigates the crack detection of FRP reinforced concrete beam using the embedded piezoelectric SA. The results indicate that the time domain amplitudes, frequency domain amplitudes and wavelet packet energies of the signal received by SA sensors decrease when the crack-induced damage occurs. The proposed method will be promising in monitoring cracks in FRP concrete structures. The paper is well written and organized. I would recommend the paper to be published in Sensors. However, the following minor issues should be addressed.

1) There are minor english language issues, it should be proofread by professional.

2) The Abstract is too lengthy. Keep it concise and should be less than 300 words.

3) The texts in Figures 1, 2, 3, 4, 6 are difficult to read. Please replace them with clear ones.

4) The following references are recommended for comprehensive review of the research topic. 

DOI: 10.1088/0964-1726/25/11/115031; 10.1177/1475921717703053

Author Response

We greatly appreciate the time and efforts by the reviewers for reviewing our paper. The insightful comments and suggestions from the reviewers greatly help improve the quality of this manuscript. All issues raised by the reviewers have been addressed in the latest manuscript, as elaborated below. For clarity, the revision in the revised manuscript are printed in blue color.

______________________________________________

Comment 1:

1. There are minor english language issues, it should be proofread by professional.

Response: The authors appreciate the reviewers suggestion. The revised manuscript has been proofread by a professional.

Comment 2:

2. The Abstract is too lengthy. Keep it concise and should be less than 300 words.

Response: The authors appreciate the reviewers careful observation and suggestion. The number of words remaining in the revised abstract is only 293, less than 300 in Paragraph 1 on Page 1.

Comment 3:

3. The texts in Figures 1, 2, 3, 4, 6 are difficult to read. Please replace them with clear ones.

Response: The authors also find that Figures 1,2, 3, 4, 6 in the peer review PDF version are difficult to read, while these in the peer review word version are clear and easy to read. Figures 1,2, 3, 4, 6 in the revised manuscript are clearly readable.

Comment 4:

4. The following references are recommended for comprehensive review of the research topic. DOI: 10.1088/0964-1726/25/11/115031; 10.1177/1475921717703053.

Response: The authors appreciate the reviewer’s suggestion. The authors refer to the two literatures recommended by the reviewer, such as [60] and [71] in Paragraph 1 on Page 3.

Reviewer 2 Report

The authors applied the simple technique of detecting accumulated cracks in reinforced concrete beam specimens during static loading. They applied smart aggregates in pitch-catch mode to register propagating elastic waves which are affected by forming cracks. The idea is that more cracks mean that the received signals are weaker. It is particularly well visible in case of applied by the author's wavelet packet-based energy analysis. The authors tested three concrete beams with different reinforcement and showed a clear trend in wavelet packet energy for each of them. The static loading test was applied from zero until the failure occurred. It would be interesting to see more complex loading and unloading scenarios in future compliant with real-world case scenarios. The critical factor would be to find the threshold level at which the structure can still be safely operated. Unfortunately, the authors did not discuss this problem.

Other than that, the paper is interesting and well written. I have only a few minor remarks:

Page 2, line 85; “to detect structural damage on an aircraft placed in a full 85 scale fatigue test “ The literature review should be directly linked to the main topic, i.e. crack detection in reinforced concrete beams. There are several unnecessary references which can be omitted.

Page 4, line 145, It should be rather “to detect the response” or “for detection of the response” instead of “for detect the wave response”

Figure 1, 2 and 4, something happened to fonts which caused that it is impossible to decipher descriptions

Page 6, line 215, “The spacing within the middle-span of about 600 mm is 100 mm” The spacing of stirrups?

Author Response

We greatly appreciate the time and efforts by the reviewers for reviewing our paper. The insightful comments and suggestions from the reviewers greatly help improve the quality of this manuscript. All issues raised by the reviewers have been addressed in the latest manuscript, as elaborated below. For clarity, the revision in the revised manuscript are printed in blue color.

______________________________________________

Comment 1:

1. Page 2, line 85; “to detect structural damage on an aircraft placed in a full 85 scale fatigue test “ The literature review should be directly linked to the main topic, i.e. crack detection in reinforced concrete beams. There are several unnecessary references which can be omitted..

Response: The authors appreciate the reviewer’s suggestion. We have omitted several references that are not relevant to the main topic in Paragraph 2 on Page 2, as shown below:

[30] Caminero , M. A.; Lopez-Pedrosa, M.; Pinna, C.; Soutis, C. Damage monitoring and analysis of composite laminates with an open hole and adhesively bonded repairs using digital image correlation. Composites Part B, 2013, 53(7), 76-91.

[33] Duncan, R. G.; Childers, B. A.; Gifford, D. K.; Pettit, D. E.; Hickson, A. W.; Brown, T. L. Distributed sensing technique for test article damage detection and monitoring. Proceedings of SPIE, 2003, 5050, 367-375.

[36] Bishop, J. A.; Pommerenke, D. J.; Chen, G. A rapid-acquisition electrical time-domain reflectometer for dynamic structure analysis. IEEE Transactions on Instrumentation and Measurement, 2011, 60(2), 655-661.

The above number is the number of the original peer review version.

Comment 2:

2. Page 4, line 145, It should be rather “to detect the response” or “for detection of the response” instead of “for detect the wave response”.

Response: The authors appreciate the reviewer’s careful observation and suggestion. We use to detect the response instead of “for detection of the response” in Paragraph 3 on Page 3.

Comment 3:

3. Figure 1, 2 and 4, something happened to fonts which caused that it is impossible to decipher descriptions.

Response: The authors appreciate the reviewer’s careful observation and suggestion. After saving the revised word version as a PDF version, Figures 1, 2, 3, 4, and 6 are now easy to read.

Comment 4:

4. Page 6, line 215, “The spacing within the middle-span of about 600 mm is 100 mm” The spacing of stirrups?.

Response: The authors appreciate the reviewer’s suggestion. The revised manuscript indicates that the spacing is the spacing of steel stirrups in Paragraph 1 on Page 6.
